# Mechanical Robust, Self-Healable Polyurethane Elastomer Enabled by Hierarchical Hydrogen Bonds and Disulfide Bonds

**DOI:** 10.3390/polym15194020

**Published:** 2023-10-08

**Authors:** Biqiang Jin, Wenqiang Wu, Haitao Wu

**Affiliations:** 1College of Science, Xichang University, Xichang 615000, China; 2State Key Laboratory of Polymer Materials Engineering, College of Polymer Science and Engineering, Sichuan University, Chengdu 610065, China; 3Sichuan Dowhon New Material Co., Ltd., Meishan 611734, China

**Keywords:** polyurethane elastomer, hierarchical hydrogen bonds, disulfide bonds, mechanical robustness, self-healing

## Abstract

The fabrication of mechanically robust and self-healing polymeric materials remains a formidable challenge. To address the drawbacks, a core strategy is proposed based on the dynamic hard domains formed by hierarchical hydrogen bonds and disulfide bonds. The dynamic hard domains dissipate considerable stress energy during stretching. Meanwhile, the synergistic effect of hierarchical hydrogen bonds and disulfide bonds greatly enhances the relaxation dynamics of the PU network chains, thus accelerating network reorganization. Therefore, this designed strategy effectively solves the inherent drawback between cohesive energy and relaxation dynamics of the PU network. As a result, the PU elastomer has excellent mechanical properties (9.9 MPa and 44.87 MJ/m^3^) and high self-healing efficiency (96.2%). This approach provides a universal but valid strategy to fabricate high-performance self-healing polymeric materials. Meanwhile, such materials can be extended to emerging fields such as flexible robotics and wearable electronics.

## 1. Introduction

Thermoplastic polyurethane (PU) elastomers are a class of elastomeric materials that can be melted by heating and have good molding and processing properties [1]. Due to their high elasticity [2], high modulus [2] and outstanding strength [3], chemical solvent and hydrolysis resistance [4] and excellent abrasion resistance [5], they have a wide range of applications in many emerging fields. However, traditional PU elastomers will inevitably encounter mechanical damage in the service lifespan [6]. Meanwhile, the inability to self-heal will lead to its products not being recycled, resulting in vast wasted resources and environmental pollution. Therefore, it is an urgent need to impart PU elastomers with self-healing ability.

Inspired by the fact that biological tissues in nature can heal themselves after damage, many researchers have explored self-healing materials from a biomimetic perspective. Self-healing PU elastomers, as a type of bionic material, have received increasing attention from researchers in recent years [7,8,9,10]. For self-healing, the current approach is mainly to release the healing agent by building microcapsules [11] or hollow micro-fiber structures [12] or rely on embedded dynamic bonds [13,14,15,16,17] to dissociate and reorganize the network under external stimuli (e.g., pH, light, heat) to facilitate healing of damaged regions of the material. Externally assisted self-healing has the drawbacks of a limited number of repairs, complicated preparation and poor regulation of mechanical properties [12]. Conversely, intrinsic self-healing can be healed unlimited times and is easy to prepare. Therefore, the current mainstream strategy for self-healing materials is intrinsic self-healing, i.e., the introduction of dynamic bonds into the polymer network. However, current polymer materials exhibit high healing efficiency at the expense of mechanical strength. This is due to high mechanical strength, and high healing efficiency arise from different molecular mechanisms.

Consequently, the simultaneous combination of high mechanical strength and high repair efficiency in an engineered elastomeric material is necessary for molecular design. The disulfide bonds in dynamic covalent bonds can effectively reduce energy input through reversible exchange reactions that can take place at a low temperature [13]. Thus, materials’ mechanical strength and self-healing efficiency are usually improved by synergistic interactions between disulfide and non-covalent bonds. For example, Zhang et al. [18] introduced uracil and aliphatic disulfide bonds into the PU backbone. Due to the synergistic effect, the fracture strength and elongation at the break of the samples reached 12.8 MPa and 442.5%, respectively, after the samples were healed at 100 °C for 12 h. Self-healing efficiency is up to 97% but needs a high healing temperature. Inspired by the work, the synergistic action of hierarchical hydrogen bonds (H-bonds) and disulfide bonds can synergistically overcome the intrinsic contradiction between the cohesion energy and relaxation dynamics of the polymer networks, leading to the PU elastomers with high mechanical strength and high healing efficiency.

In this work, we construct dynamic hard domains enriched with hierarchical H-bonds and disulfide bonds in the PU network, effectively balancing high mechanical strength and healing efficiency. Specifically, the PU elastomer was fabricated by a two-step method. Firstly, an asymmetric isophorone diisocyanate (IPDI) was reacted with a flexible polytetrahydrofuran (PTMEG) to form a prepolymer capped with isocyanate groups. Diaminodiphenyl disulfide (APD) was selected as a chain extender to provide strong H-bonds and disulfide bonds. This allows the hierarchical H-bonds and disulfide bonds formed by urethane and urea bonds to be easily introduced into the PU network. The hierarchical H-bonds and disulfide bonds act as dynamic cross-linking that can simultaneously toughen and strengthen the PU elastomer. During deformation, the strong H-bonds maintain the robustness and elasticity of the PU network, while the rupture/reorganization of the weak H-bonds and the dynamic exchange of disulfide bonds dissipate considerable stress energy.

Meanwhile, the dynamic hard domains efficiently facilitate the relaxation dynamics of the PU network chains and thus promote the healing process in the damaged regions. Therefore, the obtained PU elastomer exhibits both excellent mechanical properties (mechanical properties of 9.9 MPa, toughness of 44.87 MJ/m^3^) and high healing efficiency (96.2%). In addition, the PU elastomer shows exceptional mold and solvent recyclability. This facile but effective strategy can not only guide the design of high-performance self-healing polymers with excellent recyclability but can also be extended to a wide range of flexible electronic devices.

## 2. Results and Discussion

### 2.1. Molecular Structure and Phase Morphology

The preparation route of the target PU elastomer is shown in Figure 1. The Attenuated total reflectance (ATR) mode-Fourier transform infrared (FTIR) spectrum was recorded to investigate the molecular structure of the prepared PU elastomer, as shown in Figure 1a. The prepared PU elastomer shows negligible peaks at 2250 and 3448 cm^−1^. These peaks correspond to the stretching vibrations of N=C=O and O-H, respectively [19]. This result suggests that the isocyanate and hydroxyl groups have fully reacted to form urethane bonds. At the same time, some characteristic peaks appear on the FTIR curve at 3361 and 1545 cm^−1^, originating from the stretching and bending vibrations of the N-H group [20]. Besides, the peaks at 1720 and 1648 cm^−1^ are attributed to the C=O_urethane_ and C=O_urea_, respectively [13]. The presence of disulfide bonds is characterized by X-ray photoelectron spectroscopy (Figure 1b). These results indicate that the PU system contains disulfide bonds and hierarchical H-bonds formed by urea and urethane groups. Variable temperature FTIR spectra were recorded to further demonstrate the existence of hierarchical H-bonds in this system. As shown in Figure 1c, the peaks at 1568 and 1637 cm^−1^ gradually decrease at higher temperatures, while the peaks at 1530 and 1678 cm^−1^ increase with temperature. This result shows that the H-bonds dissociate upon heating. As a result, these results strongly support the existence of hierarchical H-bonds in this PU network.

Meanwhile, density functional theory (DFT) calculation was used to accurately calculate the bond energies of the hierarchical H-bonds from the atomic level. Small molecular modes were used to save computational resources. All molecular structures were fully optimized at the M06-2X/6-31g//B3LYP/6-31g level. The calculation of H-bonds interaction energy. The interaction energy of the H-bond is expressed by the interaction energy (Δ*E*_h_), which is the energy difference between the H-bond complex (*E*_AB_) and the monomers (*E*_A_ and *E*_B_). Firstly, optimizing the monomers (A and B) geometric structure and frequency calculation at B3LYP/6-31g level to obtain the thermodynamic energy correction value and zero-point energy correction (ZPEC). Secondly, optimizing the H-bond complex (AA_urea-urea_, BB_urethane-urethane_ and AB_urea-urethane_) and the monomers (A and B) geometric structure and frequency calculation at M06-2X/6-31g level. Meanwhile, the keyword “counterpoise” was utilized to correct the basis-set superposition error (BSSE). Therefore, the different H-bond energies are calculated:(1)ΔEh = EAB − EA − EB + BSSE
where *E*_AB_, *E*_A_, *E*_B_ and BSSE are the H-bond complex (*E*_AB_) energy, monomer A (total electronic energy), monomer B (total electronic energy) and basis-set superposition error correct value, respectively. The optimal conformations and interaction energies of H-bonds with different bond energies (urea-dimers, urea-urethane, and urethane-dimers) are shown in Figure 2a. The bond energy ranking is as follows: 13.58 kcal/mol (Δ*E*_urea-urea_) > 10.61 kcal/mol (Δ*E*_urea-urethane_) > 7.14 kcal/mol (Δ*E*_urethane-urethane_). The results of this DFT calculation demonstrate that dynamic hard domains formed by hierarchical H-bonds exist in the PU elastomer. 

PU materials show a unique phase separation structure due to the thermodynamic incompatibility of the soft and hard segments [21]. At the same time, the phase separation structure has an important influence on their mechanical and thermodynamic properties. Therefore, it is necessary to investigate the phase separation structure of the PU elastomer. Atomic force microscopy (AFM) was used to directly visualize the phase-separated structure of the PU elastomer. As shown in Figure 1d, the AFM image shows a unique two-phase structure of the PU elastomer, with the bright regions corresponding to the hard domains and the grey regions corresponding to the soft domains [22]. Thus, AFM demonstrates the existence of phase-separated structures in the PU elastomer. To further calculate the size scale of the phase region, small angle X-ray scattering (SAXS) was also used to investigate the phase separation of the PU elastomer. The SAXS curve and two-dimensional (2D) mapping image are shown in Figure 1e. There is an obvious scattering peak at 1.19 nm^−1^. Based on Bragg’s equation [23], the periodicity (d), which reflects the average distance between the hard domains, was calculated to be 5.3 nm. A schematic representation of the phase morphology of the PU elastomer is shown in Figure 1f. This unique phase behavior facilitates the mechanical properties and resilience of the PU elastomer. 

### 2.2. Thermal Properties and Network Relaxation Dynamics

The thermal properties of polymeric materials have a significant influence on their mechanical properties. Therefore, the glassy transition temperature (*T*_g_) of the PU polymer was first investigated by dynamic mechanical thermal analysis (DMA). As shown in Figure 3c, the *T*_g_ value of the PU polymer was determined by DSC to be −49.4 °C, which is attributed to the *T*_g_ of the soft chain segments. The *T*_g_ of the hard chain segments was not observed due to the dissociation of the hard domains at elevated temperatures. Collectively, the low *T*_g_ imparts the molecular chain segments with high motility at room temperature and enables rapid network reorganization, thus offering the possibility of self-healing at room temperature.

Meanwhile, the X-ray diffraction (XRD) (Figure 3a) proves the amorphous structure of the PU polymer. Therefore, the obtained PU polymer is an amorphous elastomer at room temperature. Furthermore, the DMA temperature sweep mode was performed at a heating rate of 3 °C/min from −120 °C to 120 °C to investigate the relation between storage modulus (*E*’) and temperature. As shown in Figure 3b, the *E*’ of the PU elastomer undergoes three regions of change with increasing temperature, i.e., the glassy region, the glass transition region, and the dissociation region of the dynamic binding domains region. The glassy region shows the highest *E*’ due to the frozen molecular chains; the *E*’ of the glassy transition region significantly reduces owing to the molecular chain softening. The *E*’ decreases with increasing temperature. This is due to the dissociation of the dynamic hard domains formed by hierarchical H-bonds and disulfide bonds during the heating process. 

To better investigate the effect of hierarchical hydrogen bonds and disulfide bonds on network relaxation dynamics, a stress relaxation test was performed by stretching the sample to a strain of 20% in tensile mode. The network characteristic relaxation time (τ) is defined as the time acquired for the polymer to reach 1/e (37%) of the initial stress/modulus [24]. The typical stress relaxation curves are shown in Figure 3d. The characteristic relaxation time of the PU elastomer is 282.6 s at 30 °C. This means that the topology of the network can be reorganized in a very short period, which is a prerequisite for material healing. To further investigate the effect of temperature on network relaxation, stress relaxation spectra for variable temperatures were collected (Figure 3d). As shown in Figure 3e, the network relaxation time greatly decreases as the temperature increases. Specifically, as the temperature increased from 30 °C to 60 °C, the relaxation time decreased from 282.6 s to 49.2 s. The relaxation activation energy of the network was further calculated to be 49.5 kJ/mol by fitting Arrhenius’s law (Figure 3f). Such a low activation energy for network relaxation renders it easier to heal and recycle the PU elastomer, which reduces costs and avoids some environmental pollution.

Moreover, an all-atom molecular dynamics (MD) simulation was used to theoretically illustrate the relaxation dynamics of the PU network chains (calculation details see the Appendix A). All-atom MD simulation systems are constituted by the five polymer chains, each consisting of four hard and five soft segments. The number of repeating units in the soft segments is 12. In the simulation process, the cubic box size used is 36.8 × 36.8 × 36.8 Å and three-dimensional (3D) periodic boundary conditions in three directions are used. All of the units were placed in a simulated cubic box to represent the actual molecular chain configuration, and the initial configurations were established by randomly distributing the polymer chains in simulation cells using the Amorphous Cell module of the Materials Studio (MS 2020 version). The condensed-phase optimized molecular potentials for atomistic simulation studies (COMPASSII) force field was selected to descript the interactions (cohesive energy) in the polymer chains. The amorphous cells were optimized by the Smart Minimized method. After that, the Forcite module was utilized to calculate the equilibrium configurations of polymer chains and the cohesive energy of the networks. Specifically, the cells were further annealed between 300 K and 500 K for ten circles with ten heating ramps per circle. Then, MD simulation was carried out for 300 ps NVT (keep the number of atoms, volume, and temperature constant) at 298 K and 500 ps NPT (keep the number of atoms, pressure, and temperature constant) at 298 K and 0.0001 GPa to obtain the equilibrium density. Another 500 ps NVT was carried out at a reasonable density to obtain the most stable configuration. During the calculation process, the Berendsen Barostat and Andersen Thermostat were chosen for controlling the pressure and temperature, respectively. Ewald and Atom-based were chosen as the calculation methods for electrostatic and van der Waals interactions. Figure 2b shows the snapshot of optimal conformations of the PU elastomer and their interaction energy of dynamic hard domains. To demonstrate that the dynamic hard domains formed by hierarchical H-bonds and disulfide bonds have a fast network relaxation, the interaction energy of dynamic hard domains of a comparison sample without disulfide bonds was also calculated, as shown in Figure 2c. The dynamic hard domains formed by the hierarchical H-bonds and disulfide bonds introduced into the PU network significantly reduce the interaction energy. These results reveal that the PU elastomer with dynamic hard domains shows a rapid network relaxation ability.

### 2.3. Mechanical Properties

The mechanical properties of the PU elastomer were investigated at 100 mm/min. The typical stress-strain curve is shown in Figure 4a, and the corresponding mechanical parameters are summarized in Appendix A. The PU elastomer shows outstanding mechanical performance (9.9 MPa of mechanical strength and 1424% strain at break). Such outstanding mechanical properties are attributed to the synergistic effect of hierarchical H-bonds and disulfide bonds. Aromatic disulfide bonds are susceptible to dynamic exchange at room and even lower temperatures [13]. As a result, the weak H-bonds and disulfide bonds break firstly to dissipate considerable stress energy during deformation, while the strong H-bonds maintain the robustness and elasticity of the PU network. Therefore, the elongation rupture of the PU elastomer is multi-stage, i.e., rupture of weak H-bonds and disulfide bonds and rupture of strong H-bonds. This suggests that the hierarchical designed strategy can effectively enhance the mechanical performance of the PU elastomer. Meanwhile, the PU elastomer exhibits high toughness (44.87 MJ/m^3^), comparable to some artificial materials. This excellent toughness is attributed to the rupture/recombination of hierarchical H-bonds and the dynamic exchange of disulfide bonds, resulting in significant energy dissipation.

The mechanical properties of polymer systems containing dynamic bonds are generally strain rate-responsive during deformation [14]. Therefore, the mechanical properties of the PU elastomer were investigated at different tensile speeds. As shown in Figure 4b, as the strain rate increases from 50 mm/min to 500 mm/min, the mechanical strength gradually increases, while the strain at break shows an opposite trend. This result suggests a typical rate-dependent responsive phenomenon. When the strain rate is low, the dynamic bonds have sufficient time to complete rupture/reorganization. As such, the molecular chain segments are viscous slippage in the stage, and thus, the PU elastomer shows poor mechanical strength and a large break at strain. Suppose the time scale of the strain rate increase is greater than the rupture/reorganization time scale of the dynamic bonds. In that case, the dissociated dynamic bonds will not have enough time to reorganize completely. At this point, the dynamic bonds act as dynamic cross-linkers during deformation, increasing mechanical strength while reducing the strain at break. 

### 2.4. Self-Healing Properties

Materials will inevitably undergo mechanical damage in practical applications. Therefore, it is necessary to endow polymeric materials with admirable self-healing capability. Meanwhile, it is also a promising solution to address waste of resources and environmental pollution. In the current work, introducing hierarchical H-bonds and disulfide bonds into the PU elastomer accelerates the network relaxation dynamics, thus promising their self-healing properties. To quantitatively evaluate the self-healing efficiency, the stress-strain curves of the healed specimens were measured at a tensile speed of 100 mm/min (Figure 4c). Each specimen was measured three times. Self-healing efficiency is defined as the ratio of the mechanical strength of the healed to the original specimen. At the same time, data defining self-healing efficiency in terms of elongation at break have been added to the Appendix A. The self-healing efficiencies are summarized in Figure 4d. The healed specimen shows poor mechanical strength (2.45 MPa) for healing 12 h at 25 °C; thus, the healing efficiency is only 24.7%. With time, the healed mechanical strength increases up to 4.28 MPa, 43.2% of the self-healing efficiency.

Furthermore, there is an apparent increase in healed mechanical strength as the temperature increases to 40 °C. The self-healing efficiency increases to 68.1%. As the time is prolonged, the self-healing efficiency is as high as 96.2% for 12 h healing at 60 °C. These results reveal that prolonging the healing time or increasing the healing temperature can accelerate the healing progress. The scheme of self-healing mechanism is shown in Figure 5c.

To further investigate the exchange kinetics of hierarchical H-bonds during the healing process, two-dimensional (2D) FTIR spectra were obtained from variable temperature FTIR curves. Figure 5a,b shows the synchronous and asynchronous maps of the PU elastomer. Two auto peaks at [1632, 1632 cm^−1^] and [1705, 1705 cm^−1^] are observed in the synchronous map, which is attributed to the vibration peaks of the H-bonded C=O moieties of the urea and urethane groups, respectively. The positive correlation cross peaks at [1705, 1632 cm^−1^] reveal that the carbamate and urea groups move in the same direction. Meanwhile, a positive correlation peak is also observed at the off-diagonal position of the asynchronous map. According to Noda’s rule [15] for 2D FTIR analysis, movement is preferred at this peak at 1705 cm^−1^. The results unveil that the carbamate groups move preferentially over the urea bond groups. This is because the bond energy of the hydrogen bond formed between the carbamates is lower than that of the hydrogen bond formed between the urea bonds. This experimental result of the 2D FTIR map agrees with the DFT theoretical calculations.

## 3. Experimental Section

### 3.1. Materials

Poly (tetramethylene ether glycol) (PTMEG, *M*_n_ = 2000 g/mol) was supplied by Aladdin Chemical Reagent Co., Ltd. (Shanghai, China). and was heated at 100 °C for 1 h to remove moisture before the reaction. Isophorone diisocyanate (IPDI, 99%), *p*-aminodiphenyl disulfide (APD), dibutyltin dilaurate (DBTDL, 95%), anhydrous tetrahydrofuran (THF, Safe dry 99.5%) and deionized water were purchased from Adamas (Chengdu, China). 

### 3.2. Preparation of Target PU Elastomer

The target PU elastomer was prepared by a typical two-pot polymerization procedure, as shown in Figure 1. Firstly, the reaction of PTMEG and IPDI mixed with THF was catalyzed by DBTDL at 60 °C for 12 h under an argon atmosphere to obtain the NCO-terminal prepolymer. Secondly, the chain extender, *p*-aminodiphenyl disulfide dissolved with THF solvent, was added to the prepolymer solution to react for another 24 h. Finally, the reaction solution was poured into the deionized water to precipitate the product. Meanwhile, the precipitated product was placed in the vacuum oven to obtain constant weight. The content and component information are summarized in Appendix A.

### 3.3. Fabrication of Target PU Elastomer Film

The PU elastomer film was fabricated by casting a THF solution of the PU onto a cleaned Teflon mold (100 × 100 × 10 mm). The PU sample solution was gradually dried at room temperature for 24 h, then heated at 60 °C for 12 h and the elastomer film was peeled off from the Teflon mold.

### 3.4. Characterization

Fourier transform infrared (FTIR) spectra were recorded using a Thermo Nicolet-is50 spectrometer with an attenuated total reflectance (ATR) diamond crystal at 25 °C. Variable temperature FTIR spectra were recorded in the temperature range from 25 °C to 120 °C at a heating rate of 5 °C /min. X-ray diffraction (XRD) analysis was performed on a PhilipsX’Pert PRO diffractometer (Holland) using Cu Kα radiation (λ = 0.154 nm) at 25 °C. Dynamic mechanical thermal analysis (DMA, TA Q800) was performed at a heating rate of 3 °C/min from −100 °C to 120 °C in an N_2_ atmosphere with a frequency of 1 Hz. Stress relaxation testing was carried out at different temperatures. Mechanical properties were tested using a universal testing machine (USA, Instron 5967) at a tensile speed of 100 mm/min. Each specimen was repeated three times. The self-healing test was carried out by cutting the specimen into two pieces and then recombining them for healing at different times and temperatures to quantify the healing efficiency. Scratch healing was visualized using an optical microscope (DM4P/Leica, Germany). Density functional theory (DFT) calculations. All molecular structures were fully optimized at the M06-2X/6-31g//B3LYP/6-31g level (calculation details see the Appendix A). All-atom molecular dynamics (MD) simulations were performed using the Amorphous Cell and Forcite module of the Materials Studio (MS 2020 version, Company, City, Country) to calculate the interaction energy of the PU network (calculation details see the Appendix A). 

## 4. Conclusions

We have designed and fabricated a mechanically robust self-healing PU elastomer. The core strategy is based on the construction of dynamic hard domains formed by hierarchical H-bonds and disulfide bonds. The dynamic hard domains dissipate considerable stress energy during deformation. Meanwhile, the synergistic effect of hierarchical H-bonds and disulfide bonds greatly enhances the relaxation dynamics of the PU network chains, thus accelerating network reorganization. Therefore, this strategy effectively balances the inherent contradiction between the cohesive energy and relaxation dynamics of the PU network. As a result, the PU elastomer has excellent mechanical properties (9.9 MPa and 44.87 MJ/m^3^) and high self-healing efficiency (96.2%). This designed strategy provides an excellent platform for the fabrication of high-performance, self-healing polymeric materials. Meanwhile, such polymers can be extended to emerging fields such as wearable and flexible electronic devices.

## Data Availability

The authors confirm that the data supporting the findings of this study are available within the article and its Appendix A.

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
