# Peer review of "Mechanical Robust, Self-Healable Polyurethane Elastomer Enabled by Hierarchical Hydrogen Bonds and Disulfide Bonds"

_polymers, 2023, doi:10.3390/polym15194020_

Round 1

Reviewer 1 Report

The article « Mechanical Robust, Self-Healable Polyurethane Elastomer Enabled by Hierarchical Hydrogen Bonds and Disulfide Bonds» is dealing with the development of self-healing polyurethane elastomers. To create such a polyurethane, the authors used diamino diphenyl sulfide as a chain extender. The approach to the creation of self-healing polyurethanes described by the authors is characterized by originality and novelty. All conclusions made in the article justified and supported by the results.

Nevertheless, there are several remarks to the article:

The authors state that "mechanical properties are retained at 95% after five recycling cycles", however, the article does not provide evidence for this statement.

The accompanying data file was not provided for review.

What was the thickness of the molded polyurethane elastomer film? How was the efficiency of solvent removal from the center of the film controlled?

What shape were the mechanical test specimens? What normative documents did the shape of the specimens comply with?

Author Response

My comments are

  1. The authors state that "mechanical properties are retained at 95% after five recycling cycles", however, the article does not provide evidence for this statement.

Authors’ response: Many thanks for pointing this out! Thank you very much for your careful reading of the manuscript, which was a clerical error when it was written and the author has corrected it accordingly in the text.

  1. The accompanying data file was not provided for review.

Authors’ response: Thank you very much for your careful reading of the manuscript. We have already done the uploading of the accompanying data file.

  1. What was the thickness of the molded polyurethane elastomer film? How was the efficiency of solvent removal from the center of the film controlled?

Authors’ response: Thank you for your valuable comments. The thickness of the polyurethane elastomer film is 0.5 mm. Because the thickness is uniform throughout the film, the solvent is removed together during the heating process, not from the centre in all directions.

  1. What shape were the mechanical test specimens? What normative documents did the shape of the specimens comply with?

Authors’ response: Mechanical test strips are dumbbell shaped and the dumbbell cutter meets international standards and is 4x35 mm.

Reviewer 2 Report

Dear Authors,

In this paper, the authors describe the self-healing efficiencies of polyurethane elastomer. They have thoroughly investigated the mechanical and self-healing performance, achieving excellent retention in mechanical properties after self-healing. While the authors have written the paper well, I would like to suggest adding healing efficiency measurements for elongation at break and fracture toughness values.

Author Response

My comments are
1. In this paper, the authors describe the self-healing efficiencies of polyurethane elastomer. They have thoroughly investigated the mechanical and self-healing performance, achieving excellent retention in mechanical properties after self-healing. While the authors have written the paper well, I would like to suggest adding healing efficiency measurements for elongation at break and fracture toughness values.

Authors’ response: Many thanks for pointing this out! The corresponding self-repair efficiency has been supplemented in the supporting information (Table S3).
